# Sustainability Evaluation of Municipal Solid Waste Management System for Hanoi (Vietnam)—Why to Choose the 'Waste-to-Energy' Concept

**Nguyen Huu Hoang** [1] **and Csaba Fogarassy** [2,*] 

[1] Doctoral School of Management and Business Administration, Szent István University, 2100 Gödöllő, Hungary; hoang.nguyen.huu@phd.uni-szie.hu

[2] Faculty of Economics and Social Sciences, Climate Change Economics Research Centre, Szent István University, 2100 Gödöllő, Hungary

\* Correspondence: fogarassy.csaba@gtk.szie.hu

**Abstract:** According to decision no. 491/QD-TTg signed in 2018 by the Vietnamese Prime Minister approving adjustments to the national strategy for the general management of solid waste until 2025 with a vision toward 2050, Vietnam has committed itself to move toward collecting, transporting, and treating 100% of non-household waste by 2025 and 85% of waste discharged by households by 2025. This paper aims to determine which is the best sustainable solid waste management system out of those that have been formulated by World Bank experts for Hanoi until 2030 for implementing the national strategy. The paper compares four distinct solid waste management enhancement alternatives, namely, "Improving the current system for waste collection and transportation"; "Reducing, reusing, and recycling waste at source"; "Mechanical–biological treatment (MBT) plants for classifying, composting, and refuse-derived fuel (RDF) for the cement industry"; and "MBT plants for classifying, composting, and RDF for waste-to-energy/incineration plants". The comparison was made using an analytic hierarchy process. As a result, the research indicated that "MBT plants for classifying, composting, and RDF for waste-to-energy/incineration plants" has the highest ranking in terms of a sustainable solution for the municipal solid waste management system. Therefore, it should be applied for managing the current situation in Hanoi. At the same time, the sustainable development of the system must seek to decrease the waste-to-energy ratio continuously and significantly through the planned reuse of materials that can be recycled to industry. According to the literature, in major cities in Asia and Africa, development programs are moving toward waste-to-energy solutions. The EU's circular innovation programs and action plan may be in the opposite direction to this trend.

**Keywords:** solid waste management; sustainability evaluation; MBT; MSW; AHP; analytic hierarchy process; business model innovation; waste-to-energy; circular innovation

## 1. Introduction

In recent years, municipal solid waste (MSW) management has become increasingly urgent in emerging countries due to economic growth, and the acceleration of consumption has caused an expansion in waste generation. The rise in waste creation has caused a severe shortage of landfills and higher costs for waste management [1]. MSW is also one of the public management aspects that plays a vital role in grasping opportunities and minimizing municipal and rural difficulties in relation to the negative aspects of increasing urbanization. This is a widespread issue affecting everybody in the world. The poor management of waste has been contaminating the world's oceans, clogging drains, generating floods, and transmitting infections via the breeding of vectors. Furthermore, it

also causes rises in respiratory issues through airborne particles resulting from the burning of waste, harm to creatures that consume waste unknowingly, and effects on economic development, such as reduced tourism [2,3]. Waste generation is increasing day by day; it accounts for a large amount of local budgets and government work in its treatment, and significantly affects public health as well. Waste management functions as the sole highest funding area for all administrations in low-income nations, in which it comprises approximately 20% of solid budgets, on average. In middle-income countries, solid waste management typically accounts for at least 10% of solid funds, while it represents roughly 4% in high-income nations [2]. The cost of waste management is predicted to increase by 3–4 times in developing countries, from about 20 billion US$ in 2010 to approximately 80 billion US$ in 2025. The rate of the cost increase is higher in less developed countries [3]. Therefore, finding implementable, knowledge-based solutions is significant for those countries with poor urban solid waste management performance.

Developing countries (in Africa and Asia) often have inadequate waste management systems because of limited financial resources, weak awareness, ineffective resource usage, lack of proper instruments for governance, inequality in service stipulation, overdependence on imported equipment, and sometimes improper application of technology solutions. Poor collection and disposal of urban solid waste leads to the depreciation of environmental esthetics and causes local flooding as well as land, air, and water pollution [3,4]. The consequences of these problems lead to human health hazards, which can only be lessened by implementing cost-effective technical and policy measures [5]. Many technologies have been applied to address the serious consequences of ineffective waste management that pose risks to human health and the environment. According to the hierarchy of waste management, landfilling is the most used and widespread method of solid waste (MSW) disposal worldwide [6–10]. Mechanical–biological treatment (MBT) for unsorted organic waste is one of the best and fastest technologies for the decomposition of organic components from a landfilling site [11]. Composting is a process of waste recycling based on the biological degradation of organic material under aerobic conditions, generating stabilized and sanitized compost products [12]. Of all the recycling methods, composting is recommended due to its environmental and economic benefits [12]. It has many environmental benefits, such as decreasing greenhouse gas emissions [13], minimizing leachate quantities once discarded in landfills [14], and enhancing the quality of soil when used as a soil improvement [12]. At the same time, if composting is managed and performed improperly, it may cause various environmental issues, including the formation of malodorous or toxic gases [15], dust, and bioaerosols [16,17], resulting in occupational health risks or salubrity problems to nearby residents [18]. Besides these traditional technologies, waste-to-energy technologies (WTE-T) are promising technologies, especially for developing countries, to turn waste into a useable form of energy [19]. They will play an essential role in sustainable waste management and the relief of environmental matters [20,21]. These technologies are generally classified as biological treatment technologies (or biochemical processes, such as anaerobic digestion technologies [22–27]) or as thermal treatment technologies (or thermochemical processes, such as pyrolysis [19,25,28], gasification [19,29–33], and incineration technologies [19,25,34–36]).

MSW management is a complicated and multi-dimensional issue [37]. MSW management addresses a number of factors, such as the political and legal framework, the institutional setup, the application of appropriate technologies, operations management, financial management, public participation and awareness, and the development of an action plan [38,39]. In order to manage MSW, the integration of various phases of management (sorting, collection, transport, and final destination) is paramount [40]. Many techniques, tools, and models have been applied to assess this integration and the quality of MSW management [41]. Some evaluations have been implemented based on applying sustainability indicators to assess and improve the MSW management system from different perspectives. For example, when assessing performance, including the achievement of policy goals such as "waste-aware" benchmark indicators, Chandigarh showed inferior performances in terms of environmentally controlled waste treatment, waste disposal methods, and the reduce, reuse and

recycle (3R) methodology for Surat (tier-II city) in India in comparison to other cities [42–44]. Some other studies conducted in Romania [45], in the Lombardy region of Italy [46], and in the ABC Paulista region of Brazil [47] have shown the applicability of indicators for assessing the sustainability of a MSW management system. Of all the indicators that can be applied for sustainable municipal solid waste management, it is very important to develop an appropriate policy and implementation plans to diminish the amount of waste generation by setting a system for waste separation at source and by educating citizens to raise their awareness of waste classification [48].

In 2015, the EU Commission formulated a regulatory proposal for the horizontal implementation of the Circular Economy package, which is summarized in an action program for member states. In fact, they were creating a new synonym for sustainability. The new committee, elected in 2019, has further developed this plan, which has made it a key element of the EU Green Deal Program to prioritize circular business models. The concept supports material cycling and short-cycle material transformation to save material and energy. The concept is interesting to us because this direction of development does not consider waste-to-energy utilization as part of sustainable development [49]. Many pieces of research have found that there is a difference in the interpretation of waste-to-energy programs in developed and developing countries, and even within the EU. This is because the development of circular systems requires a high level of waste selection, which is very low in developing countries, and from the background more or less missing the recycling sector. At the same time, developing countries need to expand their regional energy production processes, which can also be helped by waste-to-energy developments [50,51]. For developing countries, decision-makers prefer fast-track, cost-effective systems that have a life cycle that can be well understood and could be successful within five or a maximum of ten years [52,53].

Much research has been done to define sustainable decision-making models to evaluate waste management alternatives such as life-cycle assessment (LCA), cost–benefit analysis, and multi-criteria decision analysis (MCDA) [54]. Life cycle assessment (LCA) will support the decision-maker in choosing the best management plan with the least consequences on the environment [55]. LCA examines the environmental influence of all processes of the waste handling from "cradle to grave"; cost–benefit study analyzes the financial dimension, while multi-criteria decision analysis (MCDA) examines economic, social, and environmental criteria [56]. MCDA is often used in waste management, and this methodology is appropriate for appraising the sustainability of a waste management system. The advantage of multi-criteria analysis in evaluating the sustainable alternative is that it enables the application of both quantitative and qualitative criteria. It also permits the cooperation of various groups of decision-makers, even with contradicting intentions in determining indicators and decision-making. The conducted literature review has revealed that MCDA is quite regularly applied as a decision-making model in waste management.

In this paper, alternative methods for MSW management in Hanoi are compared based on MCDA [57]. The main aim of this article is to rank and find the most sustainable municipal solid waste management alternative for Hanoi based on waste composition and experts' opinions by using MCDA. A model based on MCDA—the AHP (analytic hierarchy process), is developed. The model is tested in a case study of the Hanoi Municipal Solid Waste Management System. Four alternatives, together with five indicators, are employed to assess the sustainability of waste treatment alternatives.

## 2. Materials and Methods

### 2.1. Research Materials

This research concentrates on analyzing the municipal solid waste management system in Hanoi, the capital of Vietnam. According to the Vietnamese General Statistics Office [58], Hanoi covers an area of about 336,000 hectares and having around 7.52 million inhabitants in 2018. The city is recorded among 17 capital cities with the biggest area globally. There are 29 administrative components at district and city level, and 584 communes, wards, and towns. Hanoi is one of the fastest-growing cities

in Vietnam. From 2015, the urbanization rate of Hanoi was 47.55%, which was 1.42 times greater than the nationwide average rate (33.40%) with an annual increase rate of 1.89%. The number of residents in urban districts was 3,699,500 persons (49.2% of their total population); while the data from rural areas was 3,823,100 individuals (50.8% of their total population). From 2018 to 2030, the urban population is projected to rise, and also the rural community will likely continue to decrease, as demonstrated in Table 1.

**Table 1.** Population and waste generation for Hanoi and forecasted data to 2030 [57].

| Item | Year 2016 | Year 2018 | Year 2030 | The Direction and Extent of the Changes in the Given Years |
|---|---|---|---|---|
| Urban population (no.) | 3,699,500 | 4,286,272 | 7,618,293 | Increasing (4–7%/year) |
| Rural population (no.) | 3,823,100 | 3,523,369 | 2,158,803 | Decreasing (4%/year) |
| Total population (no.) | 7,522,600 | 7,809,641 | 9,777,095 | Annual growth: 1.89% |
| Urban DSW generation (t/y) | 1,687,897 | 2,046,284 | 4,773,577 | Increasing |
| Rural DSW generation (t/y) | 1,144,254 | 1,103,439 | 887,366 | Decreasing |
| Total DSW generation (t/y) | 2,832,151 | 3,149,723 | 5,660,943 | Annual growth: 4.75% |
| Urban DSW gen. (kg/cap./day) | 1.25 | 1.31 | 1.72 | Increasing |
| Rural DSW gen. (kg/cap./day) | 0.82 | 0.86 | 1.13 | Increasing |
| Total DSW gen. (kg/cap./day) | 1.03 | 1.10 | 1.59 | Increasing |

DSW: Domicile Solid Waste.

The current system of waste collection, transportation, and treatment in Hanoi includes some steps as listed below.

First collection: The methods that are mainly used for collecting waste in Hanoi are wheeled bin or pushcart system, direct truck collection, and container system. Pushcarts are employed in narrow roads where waste trucks have trouble in passing. In these areas, waste collection employees push wheeled collection bins in the residential areas to accumulate solid wastes, which are set out in small (purchased) plastic bags dropped by residents along the street. Waste collection using pushcarts is implemented at least once per day, and street-sweepers clean the major streets several times every day. Therefore, in general, inhabitants are used to a top waste collection service, in which their waste is collected frequently—even when they toss it in the roads or place it in (small plastic shopping) bags at the kerbside. This system needs, nevertheless, many laborers to operate, and causes environmental issues at the transfer points.

The direct truck collection includes the small- and large-capacity vehicles with the small ones used to collect the small (shopping) plastic waste bags discarded by inhabitants on the streets and the large ones used to transport directly to landfill or treatment facilities.

The container systems are put in front of the large (e.g., high-rise) residential buildings, offices, shops, etc. for containing the waste placed by the citizens who live in these areas. After that, these wastes are collected and transported by truck to the landfill or treatment plant.

Transfer points in the streets. When the pushcarts are full, they will be placed at various vacant areas on sidewalks/pavements. The pushcarts are usually discharged directly into waste collection/transport trucks at collection points. However, when the number of pushcarts is insufficient, they are left on the ground at temporary transfer points, where the waste will remain until it is collected by a truck and is then further transported to the landfill or treatment plant. This state will lead to substantial environmental problems at the transfer points. Therefore, there is a high demand for well-planned and accurately designed, well-constructed transfer points at curbsides to put the pushcarts and containers, and empty and clean them from excess solid waste more efficiently.

Secondary collection. The small- or medium-sized compaction trucks are usually used to transfer the waste from the selected areas to the landfill/treatment facility as the secondary collection. No exclusive license is required for municipal solid waste collection, unlike the case for hazardous waste. Many collection trucks are outdated and need to be substituted by new/additional compaction trucks.

Recycling. About 10% of the municipal solid waste in Hanoi is expected to be recycled. The private and informal sectors implement most of the recycling activities. The recyclable materials and mainly packaging waste are collected and handled in the informal area before it enters the formal collection channel. Some elements are separated at source, and some other fractions are processed by workers during collection and transportation. These collectors will separate, bale, and sell waste to the processing industry. The processing of recyclable waste, to a large extent, is carried out in craft villages without any observation of operating practices. These activities pollute the air, water, and land substantially and seriously affect the health of those workers working in these areas.

Disposal/treatment. After collecting, most of the wastes are transported to Nam Son landfill for disposal. This landfill, with a total area of approximately 84 ha., is highly overloaded and needs an urgent expansion to contain the current amount of waste.

Institutional. Many companies/entities participate in waste collection, transportation, and treatment in Hanoi. There are 31 utterly independent service suppliers working to collect waste from urban and rural areas. URENCO Hanoi is accountable for gathering the waste from four downtown districts, and ten other local companies deal with the remaining urban communities, while 20 other local Joint Stock Companies collect waste out of all rural areas.

Financial. The average waste collection fee for each household in Hanoi is EUR 1.028/household/month, corresponding to EUR 0.257/person/month. This amount of fee meets only 64% of the demand for reinvestment in waste gathering and treatment activities [57].

*2.2. Alternative Solutions Compared during the Research Process*

2.2.1. Alternative 1: Improving the Current System for Waste Collection and Transportation

This alternative only focuses on improving the current selection and transportation system by utilizing transfer stations before transport to proper sanitary and fully environmentally compliant landfills. From the planning stage until 2030, in this alternative, all people who are dwelling in residential areas will have admittance to this waste selection system (100%). Several transfer stations will be established to enhance the performance of the transportation strategy. All collected waste will soon be dumped at proper landfills (Figure 1). This alternative does not assume any changes to the present informal recycling system, and the 10% recycling rate has been assumed to be maintained unchanged. The recycling rate (i.e., 10%) is an assumption at a given time and varies with time [57].

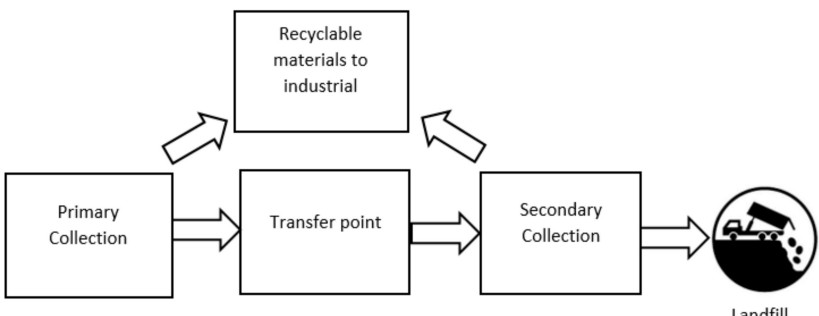

**Figure 1.** Components in Alternative 1 (Source: by authors based on World Bank survey [57]).

2.2.2. Alternative 2: Reducing, Reusing, and Recycling Waste at Source

In this alternative, the recycling rate (through collection by the informal sector) is anticipated to rise gradually from the current 10% to 24% in 2020. Moreover, Alternative 2 comprises additional sorting of recyclables at households, ranging from 1% in 2018 to 13% in 2030. Apart from sorting of recyclables at families and during the gathering and transportation procedure, the system does not incorporate any further treatment and/or reduction measures (Figure 2).

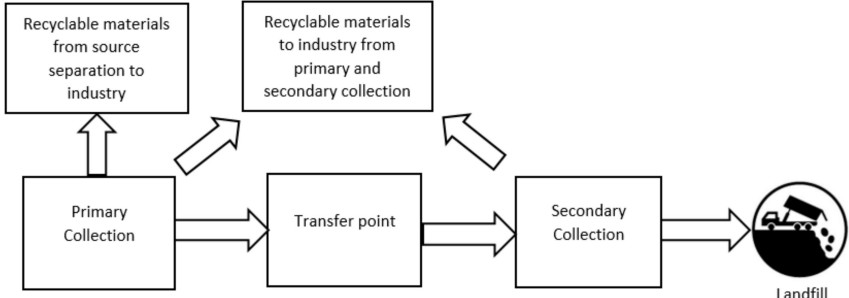

**Figure 2.** Components in Alternative 2 (Source: by authors based on World Bank survey [57]).

During the assumed planning period, until 2030, 100% of those who are living in metropolitan areas will have the possibility to use the waste collection system. More transfer stations will be established to boost the efficacy of the waste transportation system [57]. All compiled waste will be discarded at proper sanitary and fully environmentally compliant landfills.

2.2.3. Alternative 3: Mechanical–Biological Treatment (MBT) Plants for Classifying, Composting, and Refuse-Derived Fuel (RDF) for the Cement Industry

Alternative 3 comprises MBT plants for sorting, composting, and RDF for the cement industry. In this system, the MBT plants incorporate transfer stations for transferring waste using cost-effective transportation to landfills and RDF to provide for the cement industry (Figure 3). Over the assumed planning period, until 2030, all people who are dwelling in residential areas will have access to the waste collection system. The compaction trucks will be used to transport waste from the pushcarts (and containers in front of high-rise buildings) to a variety of MBT plants located at various places within the service area, thus reducing the distance and expense for transportation. After transferring to the MBT plants, waste will be sorted mechanically and manually into the following fractions:

- Materials with high-quality recyclable characteristics for the recycling industry,
- Organic substances for a compost plant located within the MBT plant (wet, small/medium particle size),
- Small unrecyclable particle fraction for landfilling such as glass, dust, soil, gravel, and
- RDF made from the remaining burnable fraction to provide for the cement industry at zero costs or to incineration facilities.

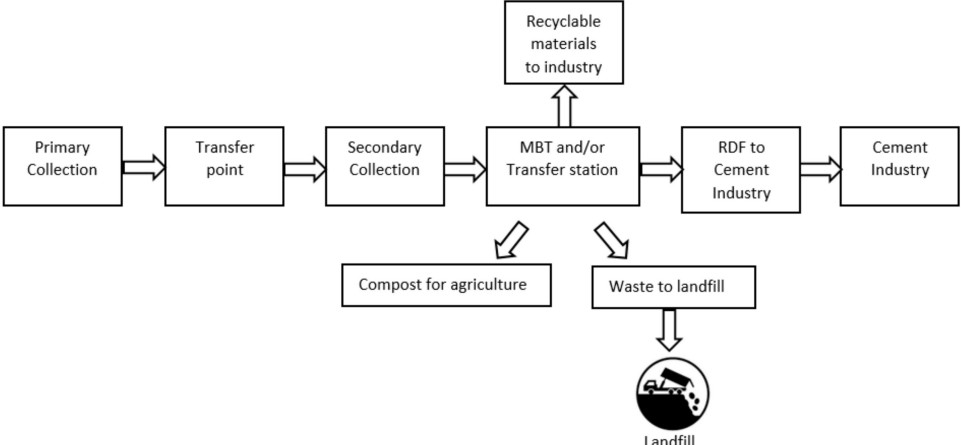

**Figure 3.** Components in Alternative 3 (Source: by authors based on World Bank survey [57]).

2.2.4. Alternative 4: Mechanical–Biological Treatment (MBT) Plants for Sorting, Composting, and Refuse-Derived Fuel (RDF) for Waste-to-Energy/Incineration Plants

Alternative 4 comprises MBT plants for classifying, composting, and RDF as fuel for waste-to-energy/incineration plants. Further, MBT plants incorporate transfer stations for the cost-effective transportation of residual waste to landfills. Over the assumed planning period, upto 2030, all people remaining in residential areas will have admittance to the waste collection system. The compaction trucks will be used to transport waste from the pushcarts (and containers in front of high-rise buildings) to a variety of MBT plants located at various places within the service area, thus reducing distance and expense for transportation (Figure 4). After transferring to the MBT plants, waste will be sorted mechanically and manually into the following fractions:

- Materials with high-quality recyclable characteristics for the recycling industry,
- Organic substance for a compost plant located within the MBT plant (wet, small/medium particle size),
- Small unrecyclable particle fraction for landfilling such as glass, dust, soil, gravel, and
- RDF from the remaining ignitable fraction. It is assumed to be burned at on-site waste-to-energy plants.

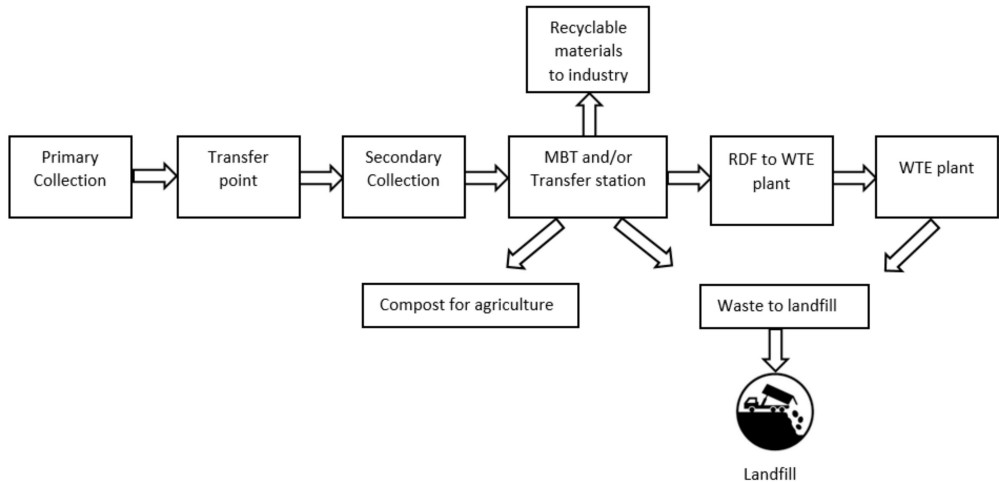

**Figure 4.** Components in Alternative 4 (Source: by authors based on World Bank survey [57]).

*2.3. Application of the Analytical Hierarchy Process (AHP)*

The AHP (analytic hierarchy process) is a multi-criteria decision-making technique, developed by Saaty [59–63]. Although AHP is an MCDA method which has been used for a long time, it is quite regularly employed to resolve complicated decision-making problems in many disciplines such as the manufacturing industry, environmental management, waste management, power and energy industry, transportation industry, construction industry, etc. In waste management, the AHP method is extensively implemented to evaluate and select the strategy for solid waste management in Bosnia and Herzegovina [64], to select the optimal strategy for solid waste management [65], or to analyze policy influence potential for solid waste management decision-making [66]. This method is also applied to determine the best alternatives for energy recovery from solid waste [67], to evaluate solid waste treatment technology, or to rank suitable solid waste facility sites [68]. Contreras et al. [69] used the AHP to select between different waste management plans to perform in Boston, USA. Substantial research has concluded that the AHP method is a strong decision tool for supporting decision-makers in the adoption of a sustainable waste management alternative. The hierarchical structure of the AHP model allows decision makers to easily understand the problems in terms of the relevant criteria and sub-criteria. Other additional criteria can be put on the hierarchical structure for

further comparison. By making the pairwise comparison between the criteria and sub-criteria with the alternatives, this model can help to prioritize and give the optimal solutions based on this information. This methodology initially requires the definition of an objective to guide the analysis, which is defined as the choice of the best alternative for sustainable MSW management in the situation of Hanoi. These alternatives are mentioned above, namely, improving the current system for waste collection and transportation; reducing, reusing, and recycling waste at source; MBT plants for classifying, composting, and refuse-derived fuel (RDF) for cement industry; and MBT plants for classifying, composting, and RDF for waste-to-energy/incineration plants. Further, criteria are defined to evaluate these options. They are: the waste flow in 2018 and forecasted data to 2030; necessary equipment and facilities from 2018 to 2030; total investments estimated for municipal solid waste collection and disposal; the annual cost of operation and maintenance for municipal solid waste collection and disposal; and the total average costs per capita per year. This research conducted an expert-roundtable. The expert-roundtable or evaluation team included two Vietnamese Ph.D. researchers with significant experience in transition management process analysis and waste management. Two experienced lead researchers of the Climate Change Economics Research Center, who have AHP practice and publications, regulatory issues, and waste management background. All three Ph.D. researchers have knowledge of waste and energy management in developing countries. During the AHP, we followed the protocol provided by the Super Decisions Software and the result was sent to the specialists in Hanoi to make the confirmation. This result included the pairwise comparisons, in which the authors had to match the criteria and sub-criteria and alternatives in pairs to assess their preferred choice. Writing the pairwise comparison matrix for Alternatives A1, A2, ... An, resulted in the matrix shown in Table 2. Alternatives: Improving the current system for waste collection and transportation (A$_1$); Reducing, reusing, and recycling waste at source (A2); MBT plants for classifying, composting, and refuse-derived fuel (RDF) for cement industry (A3); and MBT plants for classifying, composting, and RDF for waste-to-energy/incineration plants (A4).

**Table 2.** Analytical hierarchy process (AHP) pairwise comparison matrix [70].

|  | **A$_1$** | **A$_2$** | ... | **A$_n$** |
|---|---|---|---|---|
| A$_1$ | $p_1/p_1$ | $p_1/p_2$ | ... | $p_1/p_n$ |
| A$_2$ | $p_2/p_1$ | $p_2/p_2$ | ... | $p_2/p_n$ |
| ... | ... | ... | ... | ... |
| A$_n$ | $p_n/p_1$ | $p_n/p_1$ | ... | $p_n/p_n$ |

In the matrix, $a_{ij} = p_i/p_j$ shows how many times Alternative A$_{\neg i}$ is better than alternative A$_{\neg j}$ with respect to each criterion and sub-criterion.

AHP is a powerful tool used for complex decision making by enabling decision-makers to prioritize and make the decision. This methodology uses a series of paired comparisons to reduce complex decisions. By analyzing and synthesizing the results, it will help to capture both objective and subjective aspects (Figure 5). AHP is also used to lessen distortions in the decision-making procedure and constitutes a helpful way. The first step in solving decision-making tasks will be to structure the judgment task, which includes defining the goal, deciding on the alternatives, and specifying the criteria and sub-criteria.

1. Goal: Choosing the best alternative for sustainable MSW management in Hanoi;.
2. Criteria:

   a-　The waste flow in 2018 and forecasted data to 2030 (C1);
   b-　The necessary equipment and facilities from 2018 to 2030 (C2);
   c-　The total investments estimated for municipal solid waste collection and disposal (C3);
   d-　The annual cost of operation and maintenance for municipal solid waste collection and disposal (C4);

e-    The total average costs per capita per year (C5).

3.    Sub-criteria:

a-    Total of waste collection (t/y) (S1);
b-    Recycling t/y (S2);
c-    Residual waste for landfill (t/y) (S3);
d-    Transfer points in streets (no.) (S4);
e-    Pushcarts/containers (no.) (S5);
f-    Compaction trucks for collection (no.) (S6);
g-    Transfer stations (no.) (S7);
h-    New landfills needed (2 million tons capacity each) (S8);
i-    Investments estimated for collection (S9);
j-    Investments estimated for disposal (S10);
k-    Annual operation and maintenance costs for collection (S11);
l-    Annual operation and maintenance costs for disposal (S12);
m-    Total investments and reinvestments (S13);
n-    Total operation and maintenance costs (S14);

Note: Acronyms for each criterion and sub-criteria are in parentheses.

We used the "Super Decisions Software" to resolve the decision task—in this case; the AHP model consisted of the steps listed below.

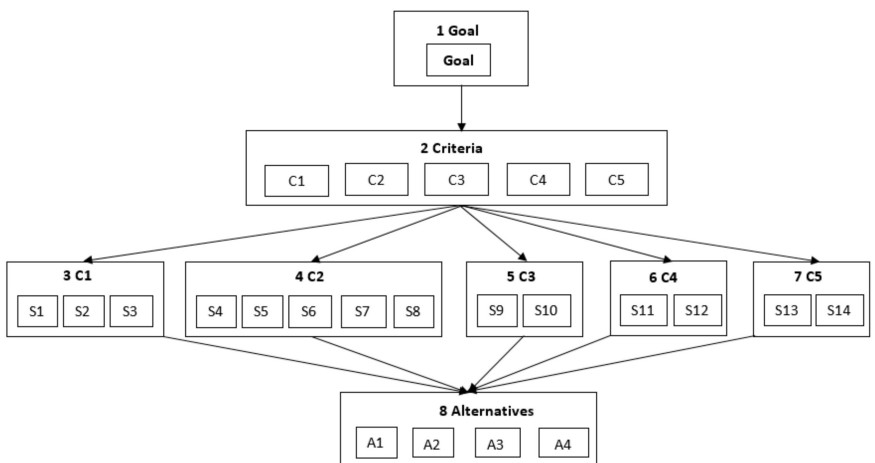

**Figure 5.** Structure of AHP analysis for sustainable municipal solid waste (MSW) management in Hanoi (Source: by authors).

Creating the Test Matrix to Determine the Weights of the Criteria and Sub-Criteria Based on the Groups of Alternatives (Table 3).

**Table 3.** Comparative matrix of individual criteria, sub-criteria, and alternatives (Based on Anita Boros and Csaba Fogarassy [70] and author's modification).

| Alternatives<br>Criteria | A1<br>($p_1 \dots p_5$) | A2<br>($p_1 \dots p_5$) | A3<br>($p_1 \dots p_5$) | A4<br>($p_1 \dots p_5$) |
|---|---|---|---|---|
| a. C1 ($a_1 \dots a_4$) | $a_1/p_1$ | $a_2/p_1$ | $a_3/p_1$ | $a_4/p_1$ |
| b. C2 ($b_1 \dots b_4$) | $b_1/p_2$ | $b_2/p_2$ | $b_3/p_2$ | $b_4/p_2$ |
| c. C3 ($c_1 \dots c_4$) | $c_1/p_3$ | $c_2/p_3$ | $c_3/p_3$ | $c_4/p_3$ |
| d. C4 ($d_1 \dots d_4$) | $d_1/p_4$ | $d_2/p_4$ | $d_3/p_4$ | $d_4/p_4$ |
| e. C5 ($e_1 \dots e_4$) | $e_1/p_5$ | $e_2/p_5$ | $e_3/p_5$ | $e_4/p_5$ |

The Evaluation Procedure of the Alternative Sccording to the Criteria Given

Saaty's major scale [62] was adopted for the comparisons between criteria and the alternatives related to each criterion. Based on the online questionnaires that were sent to the experts, they will make the paired comparison matrices to determine the weights of the decision task and evaluate the alternatives (in this case: A1, A2, A3, A4) for each leaf criteria (C1 ... C5) and sub-criteria (S1 ... S14). The scale to express the intensity of importance for each indicator is shown inTable 4.

**Table 4.** Saaty's fundamental scale [62].

| Intensity of Importance | Definition |
|---|---|
| 1 | Equal importance/preference |
| 3 | Moderate importance/preference |
| 5 | Strong importance/preference |
| 7 | Very Strong importance/preference |
| 9 | Extreme importance/preference |
| 2, 4, 6, 8 | Intermediate values of the judgment |

## 3. Results and Discussion

It is clearly seen from Figure 6 that the study determined the order of the alternatives as follows A4, A1, A2, A3, which means that Alternative A4 has the highest ranking for choosing the best sustainable municipal solid waste management system in Hanoi until 2030. In this scenario, mechanical–biological treatment facilities are applied to separate the household waste mechanically, as well as to classify the organic fraction for composing, the refuse-derived fuel fraction is then incinerated in dedicated waste-to-energy plants. This method will help to reduce the massive amount of waste that will be go to landfill and minimize the adverse effect on the environment.

This scenario will lead to an increase in recycling materials from 245,147 tons per year to about 1,068,744 tons per year in 2030; 1,045,227 tons of compost would be produced per year in 2030; and 3,285,000 tons per year of materials would be incinerated at the start (2.1 million tons per year), but this reduces to only 320,000 tons per year by 2030 [57].

The amount of waste for landfill would be reduced due to the significant positive impact from composting and the incineration in waste-to-energy plants from around 87% in 2018 to just 6% in 2030. Furthermore, the number of landfill facilities will be significantly reduced with the need of only six sites by 2030. This is an excellent effect of this scenario considering that land used for landfilling facilities in Hanoi is no longer sufficient and the negative environmental consequences from these facilities are increasing [57].

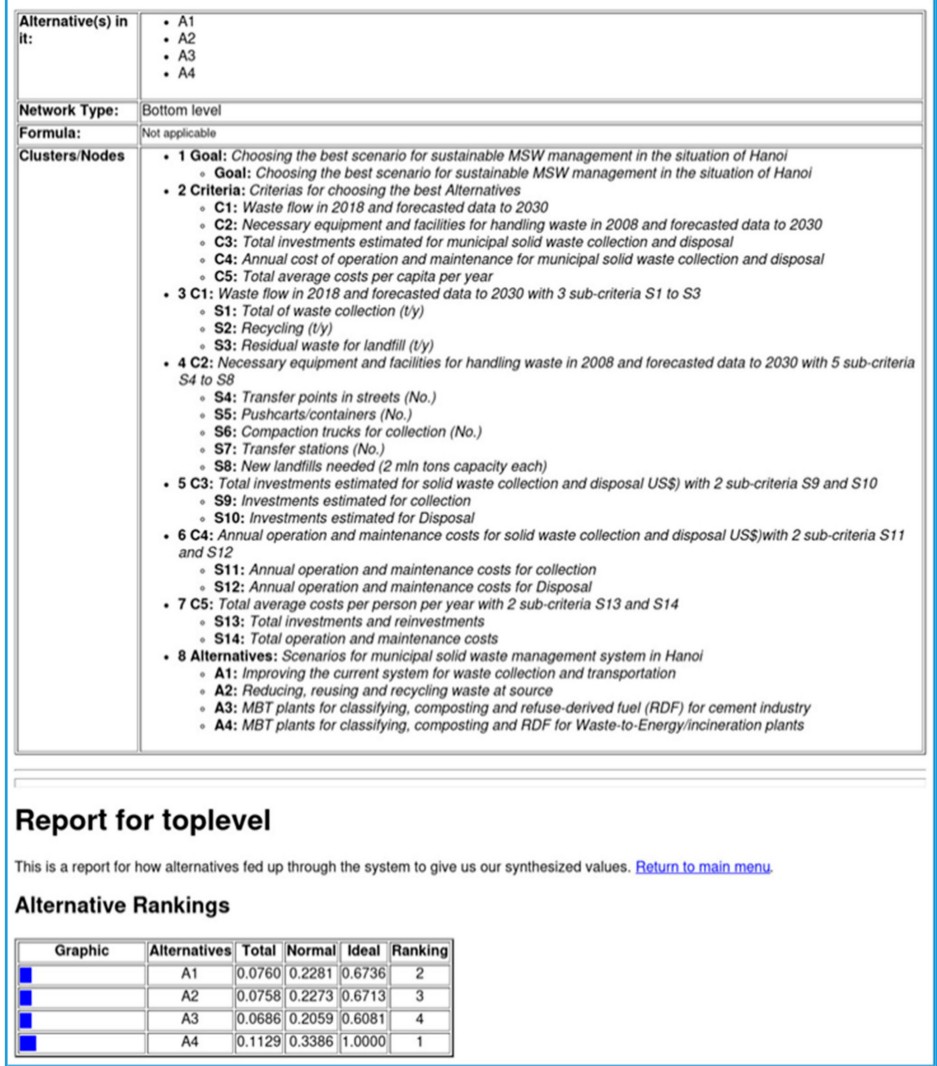

**Figure 6.** The structure of the AHP analysis and the order of the alternatives (Source: original result table from the Super Decisions Software, edited by authors).

In this scenario, the transfer stations are integrated into the MBT facilities; this means that investment in equipment and facilities needed for the modernization of the collection, transport, and disposal required in each of the four-year periods to 2030 will be considerably decreased compared to the remaining scenarios.

As in the first two alternatives (A1, A2), investments in the collection would be high in the first four years due to the supply of new pushcarts/containers and trucks. In subsequent years, the mechanical–biological treatment stations and waste-to-energy plants would be gradually rolled out, which are the main cost drivers as the new and environmentally sanitary landfill capacity required is considerably less in this alternative (only six landfills). For A1 and A2, the transformation process is slow, and it can be assumed that this type of waste management transformation while requiring the least investment, entails the highest operating costs. The system itself can lead to serious environmental and health problems because a lot of waste is dumped. In the case of Alternative A3, it has to be taken into account that the cement industry facilities have been optimized differently during their design. Poor efficiency of combustion and unfavorable greenhouse gas (GHG) emissions ranked the A3 alternative in the lowest position. In the case of A4, next to the waste collection and recycling options, the result of the cost–benefit analysis played a significant role in evaluation. The growth of employment and the growth of average household income are also important aspects of the system being set up. The result tables in

Appendix A clearly show which sub-criteria qualify for Alternative 4 (mechanical–biological treatment plants for sorting, composting, and refuse-derived fuel for waste-to-energy/incineration plants), or in which the implementation of the A4 program is neutral or unfavorable. Summarizing the details, it can be said that avoiding the construction of additional landfills, increasing waste collection and selection, and increasing the recycling rate is, according to the experts, based on the current knowledge base, the most effective solution. However, it should be emphasized that the waste-to-energy waste incineration system associated with the system solution entails higher investment and operational costs (S12, S13, S14) than the other alternatives. This is important because income levels and waste volumes are not increasing at the same rate and therefore the population is unlikely to be able to cover the costs of waste management. Higher income is associated with higher consumption, but non-environmentally friendly consumption systems generate a lot of negative externalities and waste management costs rise much faster. Compared to international data, there is a significant difference between household income and waste management costs.

The average salary in Vietnam is EUR 188.04/month. International norms indicate an affordable fee for waste management services of 1–1.5% of the average spendable income of the household. In cases where only one person generates income in the household, the affordable fee would be approx. EUR 1.879–2.819 per month/household. For the household, which has an average of four persons per household, the affordable annual fee per person would be EUR 5.645–8.468. This only covers 16–25% of the average costs, as presented above, and it is an acceptable fee for the citizens in Hanoi [57].

## 4. Conclusions

After expanding the administrative boundaries in 2008, Hanoi became one among the 17 largest capital cities in the world, with nearly 8 million people. The high proportion of urbanization and the municipal population has put enormous pressure on the infrastructure system, including the urban solid waste treatment system. Current technologies and strategies for municipal solid waste management in Hanoi are outdated, and most of the unsorted municipal wastes at source are being treated by incinerating or landfilling. This has led to severe overcrowding of existing landfills as well as causing adverse environmental impacts on air, soil, and water. Therefore, Hanoi needs a great deal in selecting and applying sustainable municipal solid waste management strategies and scenarios to minimize negative environmental impacts due to inefficient solid waste management and exploiting the energy from these waste treatment activities as well.

Development environments for waste management systems are often inadequate in developing countries. Limited financial resources, low awareness, high levels of corruption, lack of appropriate management tools, dependence on imported equipment, and inadequate technology solutions are problematic. Poor collection and disposal of urban solid waste results in aesthetic degradation of the environment and increased contamination of environmental compartments. The climatic effects (floods, fire cases, dust and air pollution, temperature extremes) as the source of dangers play in the development of appropriate technologies, in the process of adaptation. These problems also lead to human health damage, which can only be reduced by implementing cost-effective technical and policy measures. Many incorrect technologies used in waste management are posing a direct threat to human health and the environment. According to the waste management hierarchy, the most common way of disposing of MSW is untreated landfilling, which is constantly being replaced in developing countries by rapidly developing and efficient waste-to-energy systems. These can be called much safer systems, but the material and energy loss in this technology is extremely high. According to EU waste management principles, waste-to-energy technologies are not part of the sustainable development process, as all recyclable materials are lost in the technology, and only a minimal amount of energy can be realized in the process. The material and energy savings are highlighted in the EU Circular Economy Action Plan, a prominent feature of the EU's Green Deal policy. The innovation of municipal solid waste management systems in developed and developing countries is going in the opposite direction; this is because consumers have different consumption and waste selection habits.

This study evaluates the sustainability of the municipal solid waste management system in Hanoi, considering five main criteria, fourteen sub-criteria, and four alternatives by using the AHP model with Super Decisions software as a tool to make the pairwise comparison and rank the alternatives. The result of this research revealed that the scenario in which mechanical–biological treatment facilities are applied to separate the household waste mechanically, as well as to classify the organic fraction for composing, and the refuse-derived fuel fraction for incineration in dedicated waste-to-energy plants is the best alternative for the current municipal solid waste management system of Hanoi till 2030. The advantage of these scenarios is that it will help to reduce the massive volume of waste which will be landfilled and minimize the adverse effect on the environment. Furthermore, the energy produced from the waste-to-energy can be employed for many different aims.

Analytical indicators included sustainability criteria in the traditional sense but did not include circular principles. It is clear from the exploration of the European literature that development strategies of developing countries (e.g., waste-to-energy) can be sustainable in traditional interpretation (well proven in our analysis), but without circular principles, they do not facilitate the local adaptation to climate change and do not support the global climate goals either.

**Author Contributions:** Conceptualization, N.H.H.; data curation, N.H.H.; formal analysis, C.F.; investigation, N.H.H.; methodology, C.F.; supervision, C.F.; writing—original draft, N.H.H.; writing—review and editing, C.F. All authors have read and agreed to the published version of the manuscript.

**Funding:** This research was funded by the Stipendium Hungaricum Scholarship.

**Acknowledgments:** The authors would like to acknowledge the Szent Istvan University and the Stipendium Hungaricum Scholarship for supporting the research work. Special thanks to the Szent Istvan University Climate Change Economics Research Center's evaluation team for participating in the expert roundtable.

**Conflicts of Interest:** The authors declare no conflict of interest.

## Appendix A

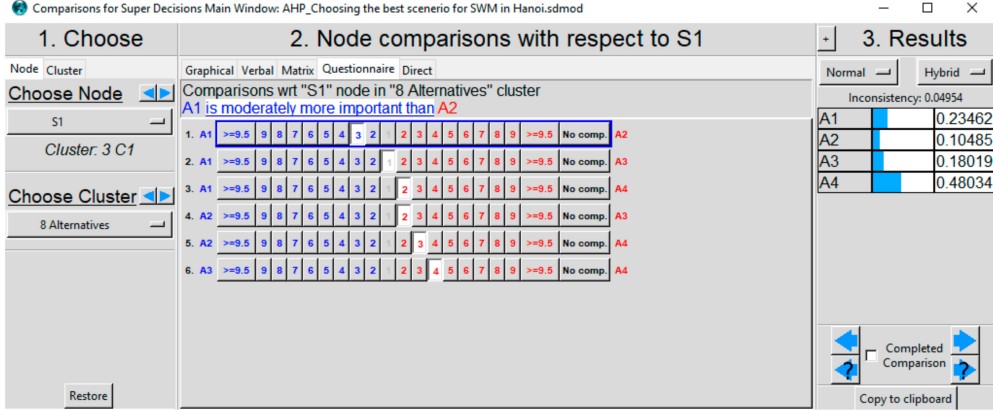

**Figure A1.** *Cont.*

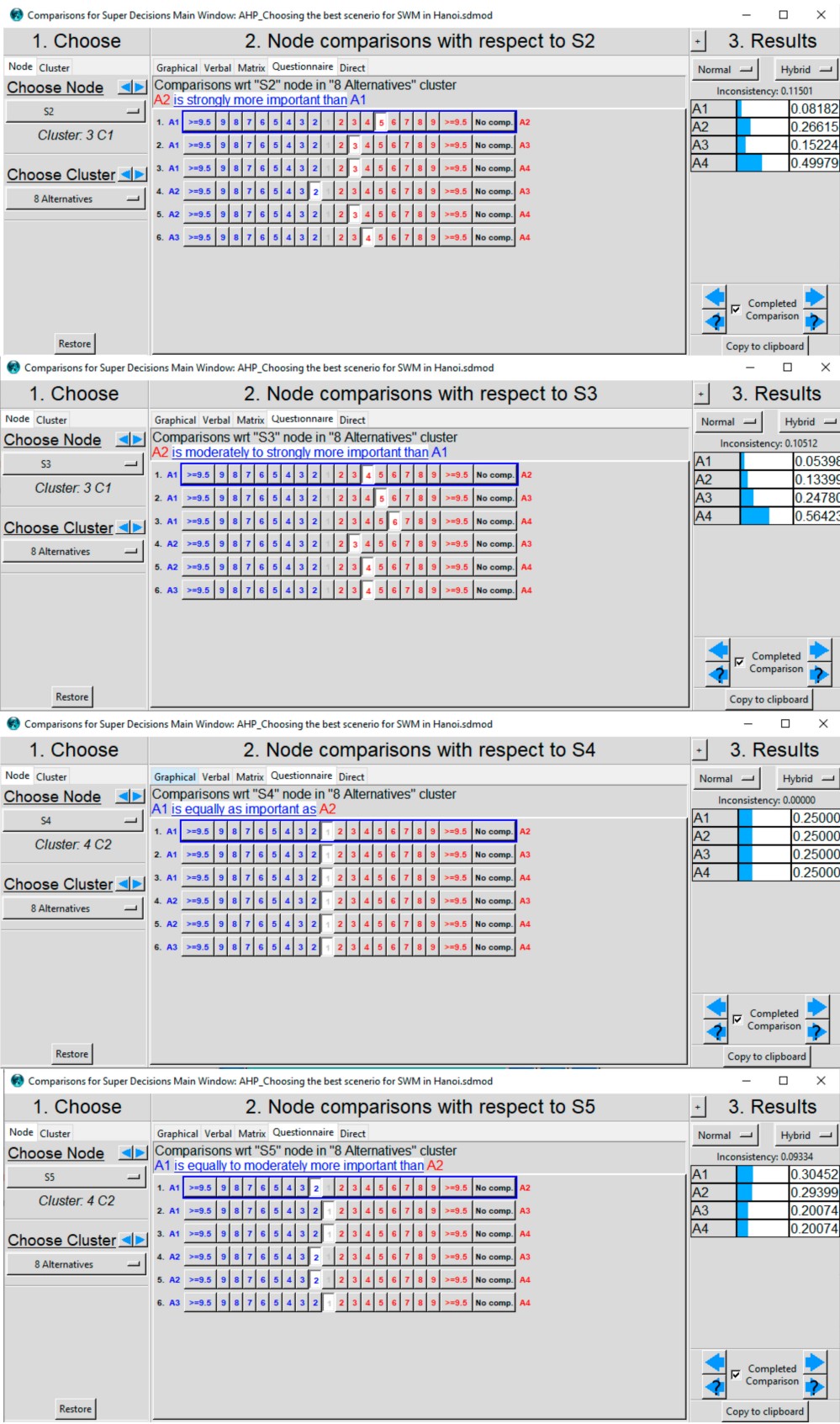

**Figure A1.** *Cont.*

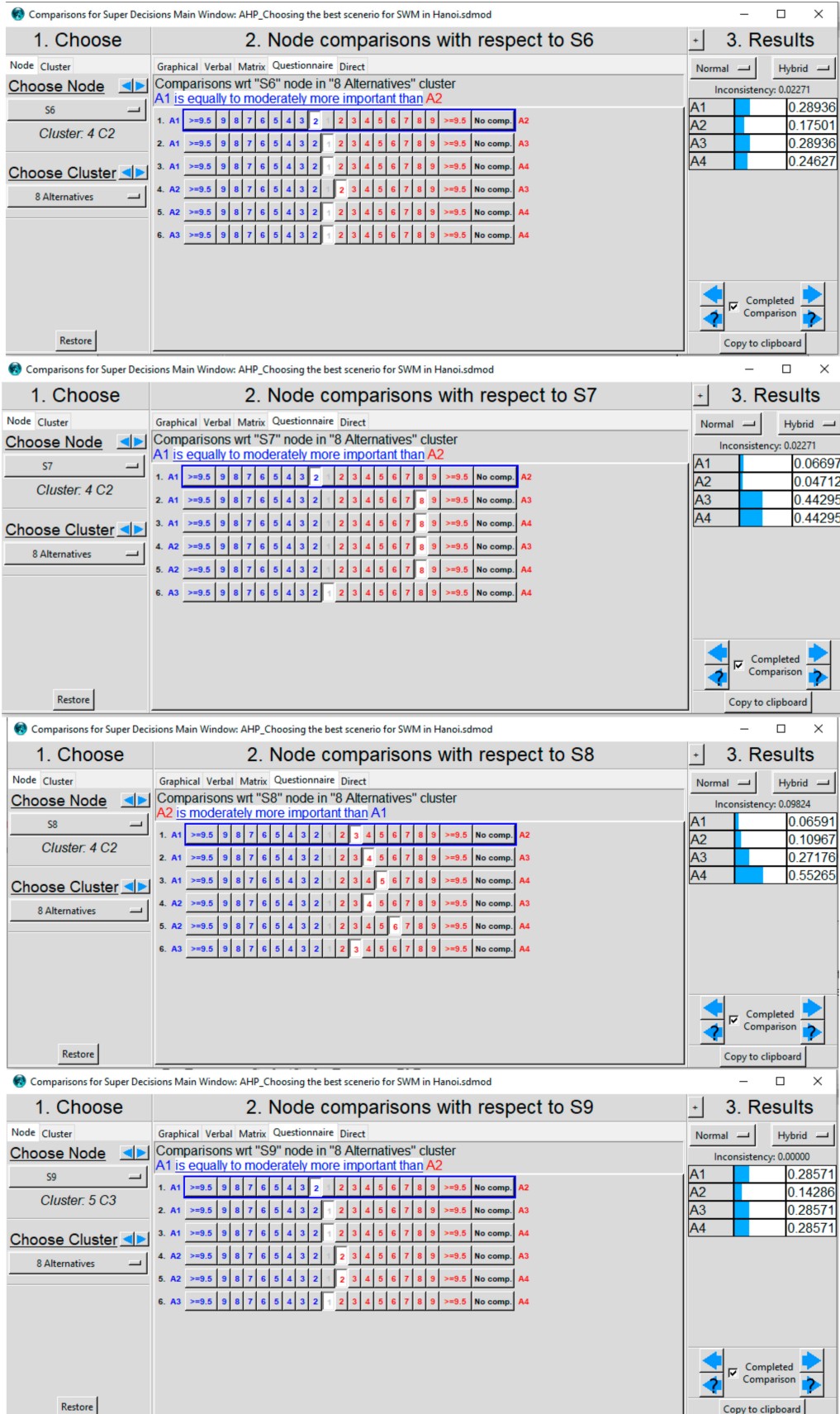

**Figure A1.** *Cont.*

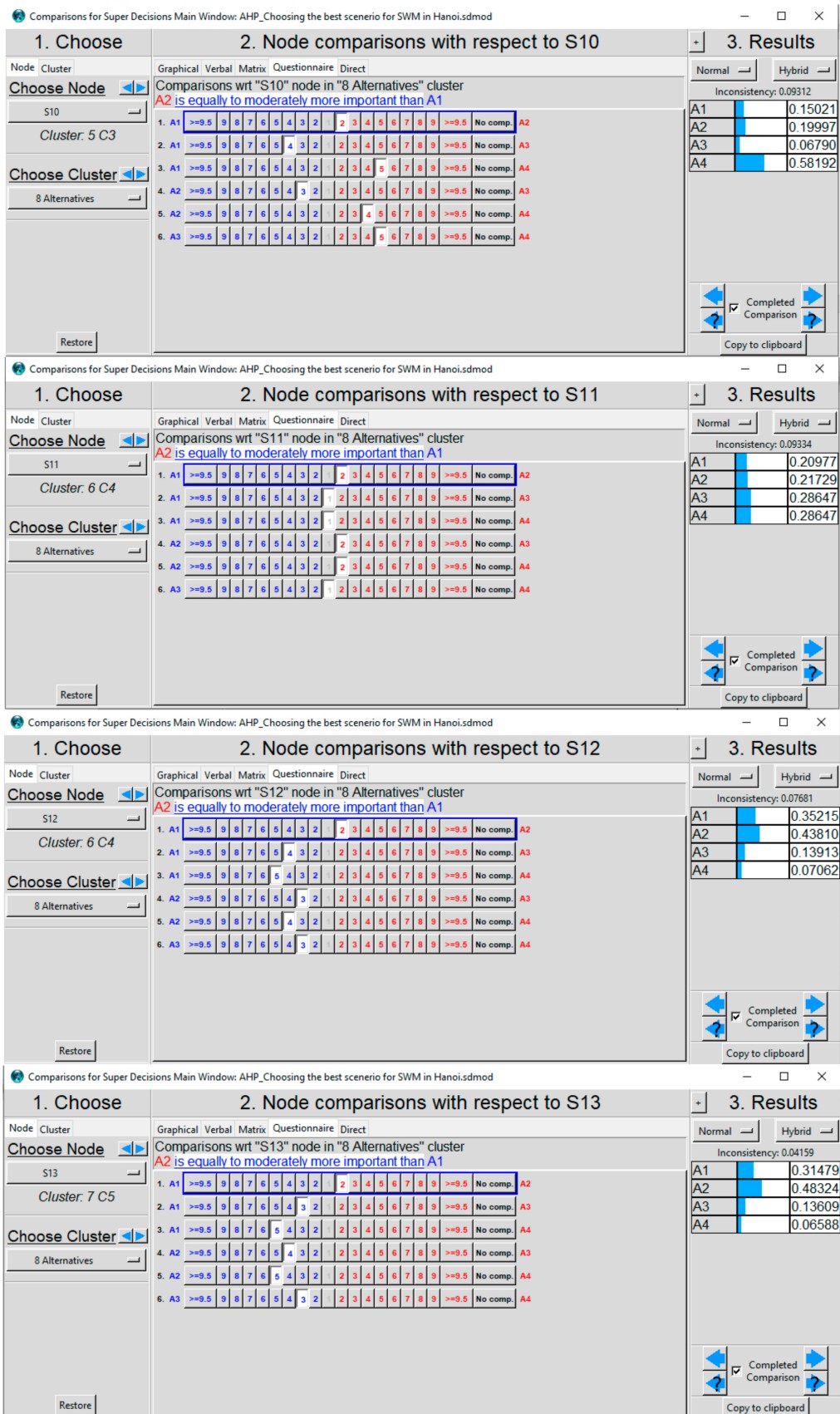

**Figure A1.** *Cont.*

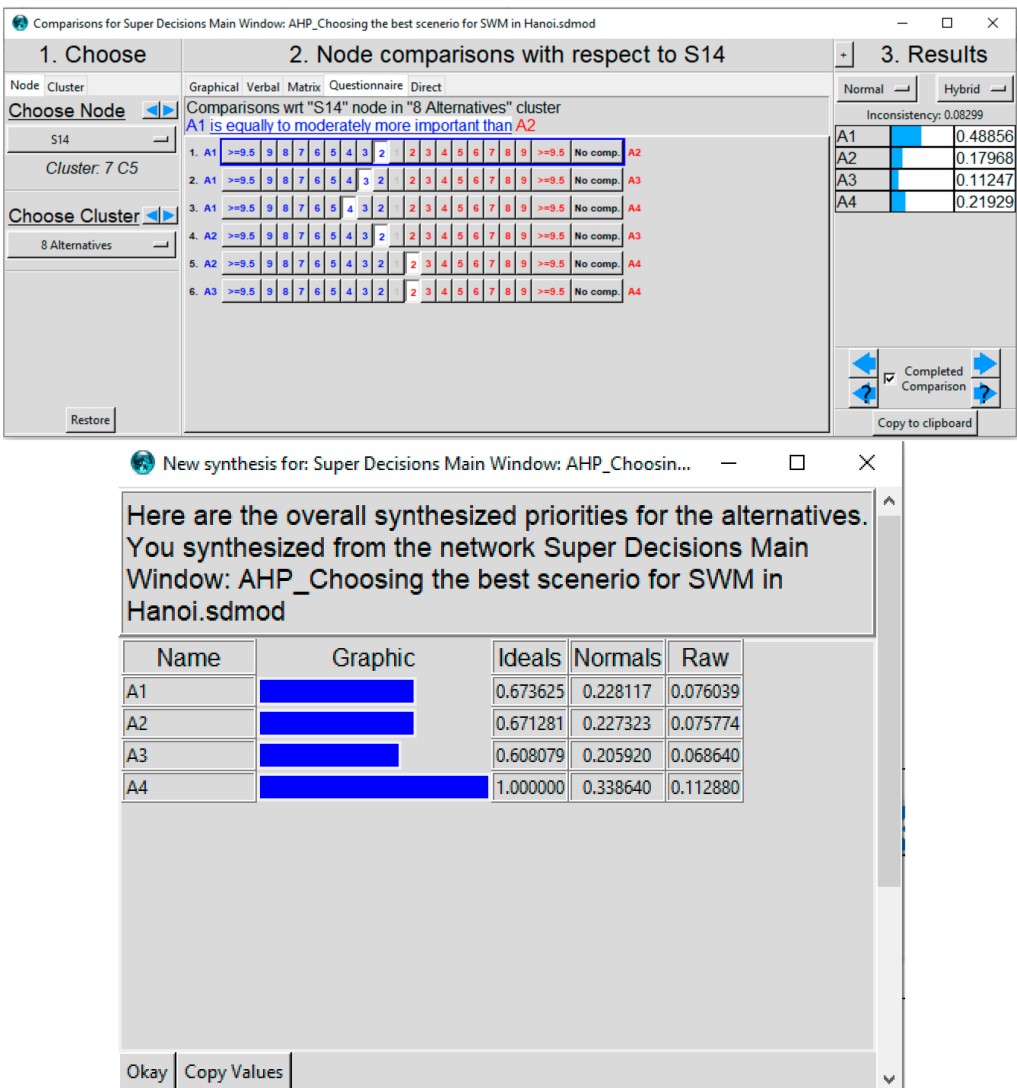

**Figure A1.** Results Chart of the Super Decisions Software Node Comparisons.

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
