# Peer review of "Sustainability Evaluation of Municipal Solid Waste Management System for Hanoi (Vietnam)—Why to Choose the ‘Waste-to-Energy’ Concept"

_sustainability, doi:10.3390/su12031085_

Round 1
Reviewer 1 Report
The paper is about the evaluation of different scenarios for MSW in Hanoi through the AHP. The topic of the paper is of potential interest, but the paper has several drawbacks.
Main issues:
the novelty of the paper is not clearly assessed and demonstrated; the methodology is unclear and confused. For example: line 219, line 220 - why these target have been set? Who decided that? Based on which data/literature? --- the authors considered that to be recycled the collected wastes need specific treatment plants? They are already available in Vietnam? The authors think they should be realized in Vietnam, or the collected waste need to be transported in foreign countries? --- line 233: use of RDF in cement industry. Is this a real potential application in Vietnam? How many cement industries there are, and where are located? --- line 244: incineration facilities. There are incinerations facilites in Vietnam? --- line 261: WTE. There is a heat demand in Hanoi, so which kind of users will use the thermal energy produced by the WTE plants? Is this a realistic case study (once again)? These scenarios are not properly justified. And why not considering anaerobic digestion in combination with MBT? online questionnaire: this part is really unclear. Please include more information on how you selected the experts, their role, expertise, number, etc... Please add the questionnaire text to the Annexes. The answers need to be analysed and clustered also in terms of people groups (technicia, politician, researcher, public administration, ...). Line 348-349: this is really unclear. Who are "they" the make the paired comparison matrices? The authors? The experts? How they make the paired comparison? Line 361-370: where these results come from? Based on which hypothesis (i.e. selection plant efficiency, recycling efficiency, etc...)
Minor issues:
the abstract is really too long. Line 35-37: unclear. line 137: unclear. line 144: repetition "Hanoi-the capital of Vietnam" line 145: "and around 7.52 millions in 2018". Inhabitants? Table 1: forecasted by who? Line 187: "about 10% ... are expected". By who? Based on which estimations?
Author Response
Dear Reviewer1,
Thank you very much for your suggestions and changes. Each comment was read and considered. All of your suggestions have been incorporated during the correction work. You can see the additions that we made!
Main issues:
the novelty of the paper is not clearly assessed and demonstrated;
Analysis of the Paper Hanoi City Waste Management Development Concept prepared by the World Bank in 2018. Four development options were compared based on sustainability indicators. These were ranked by the AHP analysis. The novelty of the result is that the 'waste-to-energy' concept is best maintained on the basis of the technologies presented. At the same time, according to the European Circular Economy Action Plan I-II, the waste-to-energy direction is not an acceptable waste management solution. The waste-to-energy concept is the winner option based on the present knowledge, but this way is not sustainable like the EU circular solutions..
the methodology is unclear and confused.
AHP and Superdecision software are well known for sorting through alternatives. In this case, four development alternatives were ranked based on the selected sustainability indicators. The pair comparisons were raised by an expert roundtable (5 experts -3 PhD students and 2 researchers), of which the authors were members. The final evaluation was made by the authors.
For example: line 219, line 220 - why these targets have been set?
In this Alternative, the recycling rate (throughout the collection by the informal sector) is anticipated to rise gradually from the current 10% to 24% in 2020. Besides, Alternative 2 comprises several additional sorting of recyclables at households, ranging from 1% in 2018 to 13% in 2030. - based on the World Bank Study report had been set on! – the data source is missing!!! – the data set on this published survey: Van Den Berg, K.D., Thuy Cam., Solid and industrial hazardous waste management assessment: options and action areas. Washington, D.C. : World Bank Group., 2018.
Who decided that? Based on which data/literature? --- the authors considered that to be recycled the collected wastes need specific treatment plants? They are already available in Vietnam?
We took the info from the World Bank study but based on the descriptions, the different units are existing. Please check the link below: https://www.cemnet.com/global-cement-report/country/vietnam
The authors think they should be realized in Vietnam, or the collected waste needs to be transported in foreign countries? --- line 233: use of RDF in the cement industry. Is this a real potential application in Vietnam? How many cement industries there are, and where are located? --- line 244: incineration facilities. There are incinerations facilities in Vietnam? --- line 261: WTE. There is a heat demand in Hanoi, so which kind of users will use the thermal energy produced by the WTE plants?
Service providers do not want to ship waste abroad, this is not one of the possible options. There is currently one cement plant in Vietnam close to Hanoi and plans to build three other plants. There is currently no incinerator or WTE. They can be built during the development process.
Is this a realistic case study (once again)? These scenarios are not properly justified. And why not considering anaerobic digestion in combination with MBT? online questionnaire: this part is really unclear.
We believe that the World Bank study is a realistic survey and a good collection of information. There was no questionnaire just online discussion with experts from Vietnam! Please check the modified description in the text!
Please include more information on how you selected the experts, their role, expertise, number, etc...
See the details of the previous answer.
Please add the questionnaire text to the Annexes. The answers need to be analysed and clustered also in terms of people groups (technical, politician, researcher, public administration, ...). Line 348-349: this is really unclear.
AHP team: two researchers and three PhD students from Vietnam, five experts with expectation on the field of waste management.
Who are "they" the make the paired comparison matrices? The authors? The experts? How they make the paired comparison?
Based on written rules that have been managed, the authors collected the info and prepare the final version of AHP.
Line 361-370: where these results come from?
This scenario will lead to an increase in recycling materials from 245,147 tons per year to about 1,068,744 tons per year in 2030; 1,045,227 tons of compost would be produced per year in 2030 and 3,285,000 tons per year of materials would be incinerated in at the start (2.1 million tons per year), but this reduces to only 320,000 tons per year in 2030 – the data set on this published survey: Van Den Berg, K.D., Thuy Cam., Solid and industrial hazardous waste management assessment: options and actions areas. Washington, D.C. : World Bank Group., 2018.
Based on which hypothesis (i.e. selection plant efficiency, recycling efficiency, etc...)
The hypothesis set on this published survey: Van Den Berg, K.D., Thuy Cam., Solid and industrial hazardous waste management assessment: options and action areas. Washington, D.C. : World Bank Group., 2018
Minor issues:
the abstract is really too long.
Yes it is true, we modified that!
Line 35-37: unclear.
„According to the literature, in the major cities of Asia and Africa, development programs are moving towards waste-to-energy solutions. The EU circular innovation programs and action plan may be in the opposite direction of this trend.” – based on the present waste management programs and literature among the references we can find this fact, because this is the cheapest and most efficient waste management solution today.
line 137: unclear.
The main aim of this article is to compare with pairwise analysis the most sustainable municipal solid waste management alternative for Hanoi based on waste composition (ranking among the alternatives), experts' opinions by using MCDA (multi-criteria decision analysis). A model based on MCDA - the AHP (analytic hierarchy process), is developed.
line 144: repetition "Hanoi-the capital of Vietnam"
line 145: "and around 7.52 millions in 2018". Inhabitants?
thank you, in both cases has been corrected!
Table 1: forecasted by who?
From the Word Bank survey ( Solid and industrial hazardous waste management assessment: options and actions areas. Washington, D.C. : World Bank Group., 2018), based on experts' opinions!
Line 187: "about 10% ... are expected". By who? Based on which estimations?
From the Word Bank survey ( Solid and industrial hazardous waste management assessment: options and actions areas. Washington, D.C. : World Bank Group., 2018), based on expert calculations!
Reviewer 2 Report
Sustainability (ISSN 2071-1050)
Manuscript ID
sustainability-694010
Title
Sustainability evaluation of municipal solid waste management system for Hanoi (Vietnam) – why to reduce the ’waste to energy’ concepts
Authors: Nguyen Huu Hoang , Csaba Fogarassy
This paper determines different alternatives (A1-A4) of various waste management technologies for MSW through the method MCDA-AHP in Hanoi. Five different criteria (C1-C5) and 14 sub-criteria (S1-S14) were chosen for the technology assessment. Research results have indicated that the alternative A4 (MBT plants for classifying, composting and RDF for WtE /incineration plants) has the highest-ranking in terms of a sustainable solution for the MSW management system. This research topic is very relevant and interesting. Nowadays the scientific community has a growing interest in relation to MCDA/LCA. Considering the properties of the most popular and commonly used MCDA-AHP method, this method can be used to contribute assessments of different technologies and scenarios and enables a clear ranking of the technologies from the point of view of their impact on the environment. Nowadays the scientific community has a growing interest in relation to MCDA/LCA. This manuscript has smaller deficiencies but the research results have good benefits in the future. MCDA and LCA complement each other well but there are still relatively few research studies combining these methods!! This topic is original, it explores several alternatives and criteria in parallel. The paper is written logically, the text clear and easy to read. The main question asked was answered, I can accept the thought process. The final conclusions are consistent with the evidence and arguments but I would continue these investigations and this topic. With these economic and social criteria, you can contribute to Life Cycle Costing and S-LCA development in the future.
Deficiencies:
Alternatives to solutions compared during the research process
Alternative 1: Improving the current system for waste collection and transportation
This part of paper I find completely okay.Alternative 2: Reducing, reusing and recycling waste at source
line 218-220: It would be important to include a literary reference that justifies the expected growth rates up to 2020 and 2030.Alternative 3: Mechanical biological treatment (MBT) plants for classifying, composting and Refuse-derived fuel (RDF) for the cement industry
Have any nearby cement factories (if any) been asked about zero costs? (line 243)Alternative 4: Mechanical biological treatment (MBT) plants for sorting, composting, and Refuse-derived fuel (RDF) for WtE/incineration plants
What and how far are the WtE technology options? Could the authors make some alternatives to this? (line 261-262) What can be highlighted within WtE opportunities? Pyrolysis? Gasification? Traditional incineration? Plasma technology?The application of Analytical Hierarchy Process (AHP)
I suggest listing alternatives earlier (line 333-336). For example, in line 207.Results and Discussions
line 361-365: It would be important to include a literary reference that justifies the expected rates in 2030.
Author Response
Dear Reviewer2,
Thank you very much for your suggestions and changes. Each comment was read and considered. All of your suggestions have been incorporated during the correction work. You can see the additions that we made!
Alternative 2: Reducing, reusing and recycling waste at source
line 218-220: It would be important to include a literary reference that justifies the expected growth rates up to 2020 and 2030.
From the Word Bank survey (Solid and industrial hazardous waste management assessment: options and actions areas. Washington, D.C. : World Bank Group., 2018), based on experts' opinions!
Alternative 3: Mechanical biological treatment (MBT) plants for classifying, composting and Refuse-derived fuel (RDF) for the cement industry
Have any nearby cement factories (if any) been asked about zero costs? (line 243)
„RDF made from the remaining burnable fraction to provide for the cement industry at zero costs or to incineration facilities.” – not necessary additional energy input into the process. .
Alternative 4: Mechanical biological treatment (MBT) plants for sorting, composting, and Refuse-derived fuel (RDF) for WtE/incineration plants
What and how far are the WtE technology options? Could the authors make some alternatives to this? (line 261-262)
What can be highlighted within WtE opportunities? Pyrolysis? Gasification? Traditional incineration? Plasma technology?
I that case the basis of the technological soultion was the Traditional incineration process!
The application of Analytical Hierarchy Process (AHP)
I suggest listing alternatives earlier (line 333-336). For example, in line 207.
Thank you for the proposal, it was removed to earlier!
Results and Discussions
line 361-365: It would be important to include a literary reference that justifies the expected rates in 2030.
Thank you for the proposal! Has been added!
Reviewer 3 Report
This paper aims to assess and choose the best sustainable solid waste management system for Hanoi and compares four distinct solid waste management enhancement alternatives. The comparison was made using an Analytic Hierarchy Process (AHP). Five different criteria were chosen for the assessment.
Section 1 is too long and it caused the motivation and purpose of this paper are not clear. In Table 1, I can not understand meaning of the row named comments. Please explain more clearly. Alternative 1, I don’t agree with the assumption that 10% recycling rate will be unchanged. Several related papers proposed that the performance of recycling could be improved in many ways. For example, people will spread the information of recycling that may improve the way of recycling. Besides, the assumption also has contradiction with the assumption of alternative 2. Please explain the way to obtain the assumption of the recycling rate in this study. The symbols in Figure 5 need to be illustrated clearly. Figure 6 needs to be remade. How many questionnaires were issued and how many experts were consulted? The result shows that the waste-to-energy waste incineration system associated with the system solution entails higher investment and operational costs than the other alternatives. In the conclusion, the author proposes that EU waste management principles “waste-to-energy technologies are not part of the sustainable development process, as all recyclable materials are lost in the technology, and only a minimal amount of energy can be realized in the process.” It seems to contradict the conclusion.Author Response
Dear Reviewer3,
Thank you very much for your suggestions and changes. Each comment was read and considered. All of your suggestions have been incorporated during the correction work. You can see the additions that we made!
Details:
Section 1 is too long and it caused the motivation and purpose of this paper are not clear.
Thank you! Has been reduced with 30 % in the corrected version!
In Table 1, I can not understand the meaning of the row named comments. Please explain more clearly.
Has been completed with more details.
Alternative 1, I don’t agree with the assumption that 10% recycling rate will be unchanged. Several related papers proposed that the performance of recycling could be improved in many ways. For example, people will spread the information of recycling that may improve the way of recycling. Besides, the assumption also has a contradiction with the assumption of alternative 2. Please explain the way to obtain the assumption of the recycling rate in this study.
These details have been calculated by the international expert from Word Bank!
The symbols in Figure 5 need to be illustrated clearly.
Thank you, has been modified in details.
Figure 6 needs to be remade. How many questionnaires were issued and how many experts were consulted? The result shows that the waste-to-energy waste incineration system associated with the system solution entails higher investment and operational costs than the other alternatives.
Thank you for your comment, has been modified!
In the conclusion, the authors propose that EU waste management principles “waste-to-energy technologies are not part of the sustainable development process, as all recyclable materials are lost in the technology, and only a minimal amount of energy can be realized in the process.” It seems to contradict the conclusion.
Very thank you for your valuable remark, we wanted to show the differences between the European innovation way and the Asian practice! We add to the Conclusion part more explanations!
Round 2
Reviewer 1 Report
The authors answered to all the reviewers' comments. Nevertheless, the doubts about novelty and methodology applied are still present.
Novelty: most (or all) of the MSW scenarios characteristics/assumptions/definitions are taken by a study of Word Bank, that was not previosly cited. So. the originality of the paper would be (only) in the comparison through AHP of these scenarios. In my opinion this is not enough to be considered as a relevant/novel scientific contribution. Methodology: the experts survey is based on only 5 experts, including 3 PhD studend, which seems to be not statistically relevant. Morevoer, it is not clear how the "expert-roundtable" has been designed and carried out. No questionnaires have been used, so how the same study can be replicated and validated?.
Author Response
Answers to Reviewer1:
Dear Reviewer1,
Thank you for your comments on the methodology and background. In an earlier, longer version, we provided the source for the original World Bank research, which, unfortunately, was omitted from the submission because of an abbreviation. About the World Bank research - an in-depth analysis by Word Bank was conducted by a large international team of experts, and we consulted with one of the team leaders during the writing of this article. We believe that the alternatives have been credibly designed, but have not been evaluated on a sustainable basis.
Regarding the originality of the paper:
The comparison includes unique indicators that have been defined by the authors or confirmed by the evaluation team. The AHP methodology is an accepted scientific methodology, the most important of which is the pairwise comparison part. The tables in the Annex give a clear description of how we have defined their relationship to each indicator. These comparisons can be updated later. The repeatability of the analysis is ensured by the established indicators and the paired comparison scoreboards.
Regarding to the Methodology:
“the experts' survey is based on only 5 experts, including 3 PhD students, which seems to be not statistically relevant. Moreover, it is not clear how the "expert-roundtable" has been designed and carried out.”
The evaluation team included two Vietnamese PhD researchers with significant experience in Transition Management Process Analysis and Waste Management. Two lead researchers have AHP practice and publications, regulatory issues and waste management background. All three PhD researchers have knowledge of waste and energy management in developing countries. During the AHP, we followed the protocol provided by Super Decision Software. In the methodological description, we added related details, describing the background associated with the AHP analysis.
Thank you very much again for your valuable suggestions. Because of your contribution to the quality and comprehensibility, our paper has been greatly improved.
The Authors
Reviewer 3 Report
In Table 1, I still can not understand the meaning of the row named “comments”. Please explain more clearly.
The recycling rate (i.e., 10%) is an assumption under a specific time point and it actually varies with time. Therefore, the recycling rate should be changed with time. If the study is under a wrong assumption, the analysis could lead to wrong results.
Figure 6 needs to be modified.
Author Response
Answers to Reviewer3:
Dear Reviewer3,
Very thank you for valuable comments, we have corrected the below mentioned issues:
In Table 1, I still can not understand the meaning of the row named “comments”. Please explain more clearly.
Answer: Has been inserted the next explanation into the Table1:
“The direction and extent of the changes in the given years”
The recycling rate (i.e., 10%) is an assumption under a specific time point and it actually varies with time. Therefore, the recycling rate should be changed with time. If the study is under a wrong assumption, the analysis could lead to wrong results.
Answer: Thank you very much for your valuable suggestion. We have added a sentence that informs the reader of the presumed data and its possible change:
"The recycling rate (i.e. 10%) is an assumption at a given time and varies with time."
Figure 6 needs to be modified.
Answer: For the sake of authentic presentation of the result, in Figure 6 we would like to leave the scoreboard copied from the evaluation software. As requested, non-informative details have now been cut off from the original Figure.
Round 3
Reviewer 1 Report
I had a quick check to the paper and I think I cannot consider it as a "new version" since very few modifications occurred from the last version. So, the remarks I expressed in my previous review remains the same, as well as my opinion about the manuscript.
Reviewer 3 Report
accept as it